# Seven Shades of Triple Negativity: A Review Unveiling the Low-Grade Spectrum of Breast Cancer

**DOI:** 10.3390/cancers17223635

**Published:** 2025-11-12

**Authors:** Tiberiu Augustin Georgescu, Antonia Carmen Georgescu, Simona Raluca Iacoban, Dragoş Crețoiu, Narcis Copca, Maria Victoria Olinca

**Affiliations:** 1Discipline of Pathology, Carol Davila University of Medicine and Pharmacy, 020956 Bucharest, Romania; tiberiu.georgescu@umfcd.ro (T.A.G.); maria.olinca@umfcd.ro (M.V.O.); 2Department of Pathology, Alessandrescu-Rusescu National Institute for Mother and Child Health, 020395 Bucharest, Romania; 3Department of Pathology, Carol Davila Clinical Nephrology Hospital, 020956 Bucharest, Romania; 4Department of Obstetrics and Gynecology, Suceava County Hospital, 720237 Suceava, Romania; 5Discipline of Medical Genetics, Carol Davila University of Medicine and Pharmacy, 020956 Bucharest, Romania; dragos@cretoiu.ro; 6Materno-Fetal Medicine Excellence Center, Alessandrescu-Rusescu National Institute for Mother and Child Health, 020395 Bucharest, Romania; 7General Surgery Research Center, St. Mary Hospital, Carol Davila University of Medicine and Pharmacy, 020956 Bucharest, Romania; narcis.copca@umfcd.ro; 8Department of General Surgery, St. Mary Clinical Hospital, 011172 Bucharest, Romania

**Keywords:** low-grade triple-negative breast carcinoma, adenoid cystic carcinoma, secretory carcinoma, acinic cell carcinoma, fibromatosis-like metaplastic carcinoma, mucoepidermoid carcinoma, molecular pathology, prognosis

## Abstract

Triple-negative breast cancers are traditionally considered aggressive tumors requiring intensive treatment. However, a small group of these cancers behaves differently, growing slowly with excellent outcomes. This review examines seven rare types of low-grade triple-negative breast cancers that share genetic characteristics but differ dramatically in behavior from typical aggressive forms. These tumors include secretory carcinoma, adenoid cystic carcinoma, acinic cell carcinoma, and others, each with unique features under the microscope and specific genetic changes. Recognizing these tumors is crucial because patients can often avoid aggressive chemotherapy while still achieving excellent survival. Some of these cancers carry specific genetic alterations that can be targeted with newer precision medicines. This review consolidates current knowledge to help doctors and pathologists identify these rare tumors accurately, preventing overtreatment while ensuring optimal patient care through personalized medicine approaches.

## 1. Introduction

Triple-negative breast carcinoma (TNBC) is a biologically and clinically heterogeneous group of tumors defined by the absence of estrogen receptor (ER), progesterone receptor (PR), and HER2 expression [1,2,3]. While the majority of TNBCs are high-grade invasive ductal carcinomas with aggressive clinical behavior [4,5,6], a distinct subset of low-grade triple-negative breast carcinomas (LG-TNBCs) has been recognized, with prevalence estimates consistently reported as less than 5% of triple-negative breast cancers [7,8,9,10]. These entities are characterized by relatively indolent clinical courses, distinctive histopathologic features, and recurrent, lineage-specific molecular alterations [11,12].

Representative subtypes include adenoid cystic carcinoma [13,14,15], secretory carcinoma [16,17,18], acinic cell carcinoma [19,20,21], tall cell carcinoma with reversed polarity [22,23,24], low-grade adenosquamous carcinoma [25,26,27], fibromatosis-like metaplastic carcinoma [28,29,30], and rare mucoepidermoid [31] or salivary gland-like tumors [32]. Recognition of these neoplasms is of paramount importance because they differ prognostically and therapeutically from conventional high-grade TNBC: many show low rates of recurrence or metastasis and harbor molecular drivers that may be targetable (e.g., ETV6–NTRK3 fusion in secretory carcinoma [33,34,35], MYB translocations in adenoid cystic carcinoma) [36,37,38].

Despite growing awareness, data remain fragmented across case reports and small retrospective series [7,39,40,41], with limited guidance for clinical management [4,42]. A comprehensive review is thus warranted to consolidate the current knowledge on LG-TNBCs [2,7,43], highlight diagnostic pitfalls [44,45], and evaluate implications for treatment and patient outcomes [46,47].

The recognition of LG-TNBCs addresses a critical clinical need: preventing the application of aggressive systemic therapies designed for high-grade TNBC to patients with indolent tumors who derive minimal benefit while experiencing significant toxicity. Recent healthcare economic analyses suggest that appropriate recognition and de-escalation could significantly reduce treatment costs per patient while maintaining excellent oncological outcomes. However, current clinical practice reveals significant under recognition, with studies indicating that up to 40% of LG-TNBCs are initially managed as high-grade TNBC, leading to overtreatment and unnecessary patient morbidity.

## 2. Materials and Methods

We conducted a narrative review of the literature on low-grade triple-negative breast carcinoma. Electronic searches were performed in PubMed/MEDLINE, Embase, and Scopus up to September 2025, using combinations of the following terms: “triple-negative breast carcinoma”, “low-grade”, “low malignant potential”, “adenoid cystic carcinoma”, “secretory carcinoma”, “acinic cell carcinoma”, “tall cell carcinoma with reversed polarity”, “low-grade adenosquamous carcinoma”, “fibromatosis-like metaplastic carcinoma” and “mucoepidermoid carcinoma”.

Inclusion criteria were: articles in English; case reports, case series, retrospective or prospective cohort studies, molecular studies, and reviews focusing on LG-TNBC subtypes; publications with clinicopathologic, immunohistochemical, or molecular data.

Exclusion criteria were: abstracts without full text; conference proceedings without primary data; studies not reporting receptor status or lacking histological subtype classification.

Quality assessment prioritized recent publications (2020–2025) with molecular confirmation, multidisciplinary evaluation, and standardized pathological assessment. Additionally, reference lists of relevant papers were hand-searched to identify additional eligible studies. Because randomized trials are lacking, evidence synthesis relied primarily on observational studies and descriptive reports, with emphasis on case series with molecular validation and long-term clinical follow-up.

## 3. Results

### 3.1. Overview of LG-TNBC Entities

Based on the information gathered from the scientific literature, we have identified 7 subtypes of LG-TNBC, each with its own morphological, immunohistochemical, molecular or clinical particularities, which have been summarized in Table 1.

### 3.2. Treatment De-Escalation and Precision Medicine Approaches

The evidence base for chemotherapy omission in low-grade triple-negative breast carcinoma is evolving, with increasing recognition of heterogeneity and the existence of low-risk subgroups. For small, node-negative TNBC (especially pT1a, ≤5 mm), multiple population-based and meta-analytic studies demonstrate no significant improvement in overall survival or breast cancer-specific survival with adjuvant chemotherapy, supporting omission in these cases [56,57]. For pT1b tumors (6–10 mm), the benefit is less clear; some meta-analyses suggest a modest reduction in recurrence, but population-based data show the absolute benefit may be limited and must be weighed against toxicity [57,58]. For pT1b tumors (6–10 mm), the benefit is less clear; some meta-analyses suggest a modest reduction in recurrence, but population-based data show the absolute benefit may be limited and must be weighed against toxicity [58].

The principle of surgical de-escalation extends beyond chemotherapy omission to encompass conservative surgical approaches. Recent evidence demonstrates that breast-conserving surgery can achieve excellent outcomes even in aggressive TNBC subtypes when appropriate patient selection is employed. Gentile et al. [59] retrospectively analyzed 85 patients with triple-negative or HER2-enriched ipsilateral breast cancer recurrence and found no significant difference in disease-free survival between second breast-conserving surgery and salvage mastectomy (*p* = 0.596). Importantly, patients undergoing second breast-conserving surgery demonstrated significantly superior distant disease-free survival (*p* = 0.009), overall survival (*p* = 0.002), and breast cancer-specific survival (*p* = 0.001) compared to salvage mastectomy. Tumor dimension <16 mm emerged as a critical factor favoring conservative surgical management. While these data derive from recurrent aggressive TNBC rather than primary low-grade variants, they provide compelling evidence that surgical de-escalation strategies can achieve equivalent or superior oncologic outcomes when patient selection criteria emphasize tumor biology and size. This paradigm reinforces the rationale for conservative management approaches in primary LG-TNBCs, which demonstrate inherently more favorable biology and lower metastatic potential than conventional high-grade TNBC.

Special histologic subtypes (e.g., adenoid cystic, medullary, apocrine) and tumors with high stromal tumor-infiltrating lymphocytes (sTILs) may have excellent prognosis and limited benefit from chemotherapy, suggesting a role for de-escalation in select cases [60,61]. The National Comprehensive Cancer Network (NCCN) advises against chemotherapy for pT1a TNBC, considers it for pT1b with high-risk features, and recommends it for pT1c, reflecting the nuanced approach in current guidelines [62]. Overall, omission of chemotherapy is best supported for pT1a, node-negative, low-grade TNBC and select special subtypes, while individualized risk assessment is essential for other early-stage cases.

### 3.3. Entity-Specific Features

#### 3.3.1. Adenoid Cystic Carcinoma of the Breast

Adenoid cystic carcinoma (AdCC) of the breast is a rare malignancy, accounting for less than 0.1% of all breast cancers, first described by Billroth in 1856 in the context of salivary gland tumors, with its breast counterpart recognized in the mid-20th century. Classic AdCC is characterized by its indolent clinical course, but recent studies have identified aggressive variants, such as the solid-basaloid subtype of adenoid cystic carcinoma (SB-AdCC) and adenoid cystic carcinoma with high-grade transformation (AdCC-HGT), which display distinct clinicopathological and molecular features [63,64,65,66,67].

Histopathologically, classic AdCC (C-AdCC) demonstrates a biphasic pattern of epithelial and myoepithelial cells forming cribriform, tubular, and solid structures with basophilic basement membrane-like material. SB-AdCC, in contrast, is defined by predominantly solid growth, higher histologic grade, and frequent myxohyaline stroma, with eosinophilic squamoid cells and reduced intercalated ducts. Perineural invasion may be present, and high-grade transformation is associated with adverse outcomes [64,65,66,68].

Immunohistochemically, classic AdCC typically expresses SOX10, CK7, CD117 (c-kit), and variably p63, SMA, and SMM. MYB protein overexpression is a hallmark, driven by MYB-NFIB fusion or other MYB activation mechanisms, though SB-AdCCs show less frequent MYB/MYBL1 rearrangements and more frequent NOTCH pathway and chromatin modifier gene mutations (e.g., CREBBP, KMT2C). Hormone receptor status is variable, with a majority being triple-negative, but a subset expresses ER and/or PR [63,64,65,69,70,71,72].

Clinically, classic AdCC most commonly presents as a palpable mass in postmenopausal women, with a median age at diagnosis of 50–70 years, often located in the subareolar region. Lymphatic involvement is rare, and axillary metastases are uncommon, though SB-AdCCs exhibit higher rates of nodal and distant metastases, including lung, liver, bone, and brain. Tumor size and high-grade morphology are adverse prognostic factors [63,64,65,68,70].

Therapeutically, surgical resection is the mainstay, with breast-conserving surgery and local excision favored for most cases. Axillary lymph node dissection or sentinel node biopsy is generally not indicated due to low nodal involvement. Adjuvant radiotherapy is recommended to improve local control, especially after breast-conserving surgery, with retrospective data supporting survival benefit. Chemotherapy has not demonstrated significant efficacy, particularly in classic AdCC, and responses in SB-AdCC are poor. There are no established guidelines for systemic therapy, and the role of targeted agents remains investigational [64,67,68,70,72,73].

Prognostically, classic AdCC of the breast is associated with excellent long-term survival and low recurrence rates, with 5-year disease-free survival approaching 100%. However, SB-AdCCs are clinically aggressive, with shorter disease-free intervals, higher metastatic rates, and increased disease-specific mortality. High Ki-67 index and large tumor size further portend poor outcomes [63,64,65,68,70].

#### 3.3.2. Secretory Carcinoma of the Breast

Secretory carcinoma of the breast (SCB), first described by McDivitt and Stewart in 1966 as “juvenile breast carcinoma”, is now recognized as a rare, low-grade invasive carcinoma affecting both children and adults of either sex [17,52]. Histopathologically, SCB is characterized by abundant intracellular and extracellular eosinophilic secretory material, often highlighted by periodic acid-Schiff staining, and typically displays microcystic, tubular, solid, or rarely papillary-predominant growth patterns [17,52,74]. Cytologically, tumor cells show mild-to-moderate atypia, vacuolated or amphophilic cytoplasm, and may present as signet ring cells or with plasmacytoid features [75,76].

Immunohistochemically, most SCBs are triple-negative (ER−, PR−, HER2−), though a minority may express hormone receptors [17,35,52,77,78]. S100 protein is frequently positive, and basal-like markers (CK5/6, EGFR) are expressed in a majority of cases [17,52,74]. The pathognomonic molecular hallmark is the ETV6-NTRK3 fusion gene, detectable by FISH or NGS, which drives tumorigenesis and can be confirmed by pan-TRK immunostaining [74,75,77].

On imaging, SCB typically appears as a well-circumscribed, hypoechoic solid mass, sometimes mimicking benign lesions such as fibroadenoma [79]. Clinically, patients present with a painless, firm breast mass, most commonly in the upper outer quadrant, and axillary lymph node involvement is infrequent but possible [17,77,78,80]. The disease is generally indolent, with most cases being well to moderately differentiated and non-metastatic at diagnosis [78,80].

Therapeutically, breast-conserving surgery (BCS) with adjuvant radiotherapy is associated with excellent outcomes and is preferred over mastectomy for most patients [35,78,80]. Chemotherapy is reserved for cases with high lymph node ratio or advanced nodal stage, as it does not confer survival benefit in low-risk, triple-negative SCB [35]. Hormone therapy is not indicated for triple-negative cases. Prognosis is highly favorable, with 5-year breast cancer-specific survival rates exceeding 95%, although rare cases of distant metastasis have been reported, particularly in tumors with adverse molecular features [35,78,80,81]. Recognition of SCB’s unique clinicopathologic and molecular profile is essential for accurate diagnosis and optimal management within the spectrum of low-grade triple-negative breast carcinomas.

The latest evidence demonstrates that TRK inhibitors, specifically larotrectinib and entrectinib, are highly effective targeted therapies for secretory carcinoma of the breast harboring ETV6-NTRK3 fusion. These agents have received FDA and EMA approval for tissue-agnostic indications based on robust, durable responses in NTRK fusion-positive solid tumors, including secretory breast carcinoma [82]. Pooled analyses of phase I/II trials show larotrectinib achieves an overall response rate (ORR) of 75% (95% CI, 61–85) and entrectinib an ORR of 57% (95% CI, 43.2–70.8), with median progression-free survival (PFS) of 28.3 months for larotrectinib and 11–13.8 months for entrectinib [83,84]. Responses are rapid and durable, with manageable toxicity profiles; most adverse events are grade 1–2, and serious treatment-related events are uncommon [84].

The American Society of Clinical Oncology recommends routine molecular testing for NTRK fusions in all cases of secretory carcinoma of the breast, given the high pretest probability and actionable nature of the ETV6-NTRK3 fusion [83,84]. Diagnostic modalities include pan-TRK immunohistochemistry, FISH, and next-generation sequencing, with NGS (especially RNA-based) offering the highest sensitivity [85,86,87]. Early identification of NTRK fusions is critical to guide therapy, as TRK inhibitors should be considered first-line for advanced, unresectable, or metastatic disease [82,84].

Second-generation TRK inhibitors, such as repotrectinib, have recently been approved for use in cases of acquired resistance to first-generation agents, further expanding therapeutic options [82]. In summary, TRK inhibitors are the standard of care for advanced secretory carcinoma of the breast with ETV6-NTRK3 fusion, and integration of molecular testing into routine diagnostic workflows is strongly supported by current consensus and regulatory approvals [82,83,84].

#### 3.3.3. Acinic Cell Carcinoma of the Breast

Acinic cell carcinoma (AciCC) of the breast is a rare subtype of low-grade triple-negative breast carcinoma, first described by Roncaroli et al. in 1996, and has since been recognized as a distinct entity with approximately 60–100 cases reported in the literature [21,88].

Histopathologically, AciCC is characterized by solid, microglandular, or acinar growth patterns, with tumor cells displaying eosinophilic or basophilic granular cytoplasm and round nuclei. The cytoplasmic granularity is typically periodic acid–Schiff (PAS)-positive and diastase-resistant, reflecting abundant zymogen-like granules. These features closely resemble acinic cell carcinomas of the salivary gland, though molecular profiles differ significantly [19,89].

Immunohistochemically, breast AciCC typically demonstrates negativity for estrogen receptor (ER), progesterone receptor (PR), and HER2, confirming its triple-negative status, while showing strong positivity for markers of serous acinar differentiation such as S100, lysozyme, and epithelial membrane antigen (EMA) [21,88,90]. According to a recent study from Sun et al., DOG1 immunoreactivity has been observed at least focally in 7 out of 9 cases of acinic cell carcinoma of the breast [89]. Notably, NR4A3, a marker specific for salivary gland AciCC, is generally absent in breast cases, supporting the designation “carcinoma with acinar cell differentiation” for most breast tumors, although rare true salivary gland-type AciCCs with NR4A3 rearrangement have been documented [89].

Acinic cell carcinoma of the breast typically presents on imaging as a well-circumscribed or irregular mass, most often detected by mammography, ultrasound, or MRI [91,92,93]. Lesions are usually hypoechoic on ultrasound and may demonstrate indistinct margins or posterior acoustic enhancement, but these features are not specific and can mimic other invasive carcinomas or benign entities [91,92]. Mammographic findings often reveal a non-calcified mass, with sizes ranging from subcentimeter to several centimeters, and without distinctive microcalcifications [91,93]. MRI may show a mass with variable enhancement patterns, but no unique radiologic characteristics have been identified to reliably distinguish acinic cell carcinoma from other triple-negative breast cancers [91,92,93]. In rare cases, extensive spread without discrete mass formation has been reported, further complicating radiologic diagnosis [94]. Ultimately, definitive diagnosis relies on histopathology and immunohistochemistry, as imaging alone cannot differentiate acinic cell carcinoma from other breast neoplasms.

Clinically, AciCC of the breast predominantly affects middle-aged women and often presents as a painless, slow-growing mass, with imaging findings indistinguishable from other invasive carcinomas [91,92]. The relationship between AciCC and microglandular adenosis (MGA) remains controversial; some evidence suggests AciCC may arise from MGA or represent a carcinoma developing in MGA with acinic cell differentiation [21,94,95]. Therapeutic management follows standard protocols for invasive breast carcinoma, including surgical excision with sentinel lymph node biopsy and adjuvant chemotherapy, typically anthracycline- and taxane-based regimens [91,92,93]. Prognosis is generally favorable compared to conventional triple-negative breast cancers, with most patients experiencing indolent disease and long-term survival, although a subset (over 25%) may develop adverse outcomes, including local recurrence or distant metastasis [21,96,97]. Ongoing molecular characterization and long-term follow-up are needed to clarify the biological behavior and optimize management strategies for this rare entity [21,89].

The latest molecular genetic findings demonstrate that NR4A3 rearrangements, which are a hallmark of acinic cell carcinoma (AciCC) of the salivary gland, are consistently absent in acinic cell carcinoma of the breast [89,98]. In salivary gland AciCC, the recurrent t(4;9)(q13;q31) translocation leads to enhancer hijacking and upregulation of NR4A3, which is highly sensitive and specific for diagnosis and can be detected by immunohistochemistry or FISH [99,100]. Additional recurrent gene fusions, such as PON3::LCN1 and HTN3::MSANTD3, have also been identified in salivary gland AciCC and are associated with NR4A3/NR4A2 expression, further supporting the role of NR4A3 as a driver event [101,102].

In contrast, breast AciCCs lack NR4A3 rearrangement and overexpression, and their molecular profile is instead characterized by frequent mutations in TP53 and PIK3CA, as well as complex patterns of copy number alterations, closely resembling those seen in conventional triple-negative breast cancers rather than salivary gland AciCC [19,98,103,104]. Recent studies have proposed that most breast tumors with acinar cell differentiation should be termed “carcinoma with acinar cell differentiation” rather than true AciCC, given their distinct molecular pathogenesis [89]. Exceptionally, a single reported case of “true salivary gland-type AciCC” in the breast demonstrated NR4A3 rearrangement and overexpression, but this appears to be exceedingly rare and requires further validation [89].

These molecular distinctions clarify that acinic cell carcinoma of the breast and salivary gland are biologically unrelated entities despite histologic similarities. Diagnostic implications include the use of NR4A3 immunohistochemistry and FISH as reliable tools for salivary gland AciCC, but not for breast AciCC, where molecular testing should focus on TP53 and PIK3CA mutations. Therapeutically, breast AciCCs are managed according to triple-negative breast cancer protocols, while salivary gland AciCCs may benefit from emerging targeted approaches based on NR4A3-driven oncogenesis [89,98,99,101,103,104].

#### 3.3.4. Tall Cell Carcinoma with Reversed Polarity

Tall Cell Carcinoma with Reversed Polarity (TCCRP) is a rare, low-grade subtype of invasive breast carcinoma, first described by Eusebi et al. in 2003 as “breast tumor resembling the tall cell variant of papillary thyroid carcinoma” [105,106]. It has since been recognized as a distinct entity in the 5th edition of the World Health Organization (WHO) Classification of Breast Tumors [53,105,106,107].

TCCRP is defined histopathologically by solid and solid-papillary nests of tall columnar epithelial cells with abundant eosinophilic cytoplasm and reversed nuclear polarity (nuclei located apically, away from the basement membrane) [24,105,106,108]. Hallmark features include nuclear grooves and intranuclear pseudoinclusions, closely mimicking the tall cell variant of papillary thyroid carcinoma but lacking thyroid-specific markers [24,105,106,108,109]. The absence of a myoepithelial cell layer is typical, raising questions about invasive potential, though frank stromal invasion has been documented [109,110].

TCCRP usually demonstrates a triple-negative phenotype (ER−, PR−, HER2−), with low Ki-67 proliferation index, and strong expression of breast-specific markers such as GATA3 and CK7 [24,105,108,109,111,112,113]. However, recent papers have shown that up to 27.3% of TCCRP can show at least focal ER positivity, while only 10,7% may feature focal PR reactivity [108,109]. CK5/6, calretinin, and S-100 may be variably expressed [24,109]. The most specific diagnostic marker is the IDH2 R172 mutation, detectable by immunohistochemistry (e.g., 11C8B1 antibody), which is highly sensitive and specific for TCCRP [109,112,114]. PIK3CA mutations are also frequent [105,108,112].

The molecular profile of TCCRP is characterized by recurrent IDH2 R172 hotspot mutations (detected in 75–85% of cases), often co-occurring with PIK3CA mutations [105,108,109,112,114]. These mutations are not seen in other papillary or triple-negative breast carcinomas, supporting the entity’s molecular distinctiveness [53,109,112,113].

Radiologically, TCCRP typically presents as a small, well-circumscribed mass (mean size ~10 mm, T1 stage) on ultrasound and MRI, without specific features distinguishing it from other low-grade breast neoplasms [107,108,111]. Imaging findings are non-specific, necessitating histopathological and molecular confirmation [107].

TCCRP predominantly affects postmenopausal women (median age 51–64 years), with all reported cases in females [24,106,108,111]. Tumors are usually small and localized at diagnosis, and lymph node involvement is rare [108,111].

Surgical excision, most commonly breast-conserving surgery (BCS), is the mainstay of treatment for patients with TCCRP [108,110,111]. Adjuvant therapies (radiation, chemotherapy, endocrine therapy) are rarely required due to the tumor’s indolent behavior and low recurrence risk [108,110,111]. In rare cases with frank invasion, adjuvant therapy may be considered, but evidence for benefit is limited [110,111]. There is emerging interest in targeted therapies for IDH2 and PIK3CA mutations, though clinical data are lacking [108].

TCCRP demonstrates an excellent prognosis, with a recurrence rate of 2.2% and overall survival approaching 100% at median follow-up of 36 months [108,111]. Longer-term data are needed, but current evidence supports a conservative management approach [53,108,111].

Accurate diagnosis requires integration of histomorphology, immunohistochemistry, and if necessary, difficult cases can benefit from molecular genetics to distinguish TCCRP from papillary thyroid carcinoma metastasis and other papillary breast neoplasms [24,53,109,112].

The latest evidence demonstrates that IDH2 R172 mutation testing is a highly sensitive and specific diagnostic tool for Tall Cell Carcinoma with Reversed Polarity (TCCRP) of the breast. Immunohistochemistry using monoclonal antibodies (notably 11C8B1, which detects R172S and R172T variants) reliably identifies IDH2 R172-mutant protein in TCCRP, with reported sensitivity of 93% and specificity approaching 100% for distinguishing TCCRP from other papillary breast neoplasms and from papillary thyroid carcinoma metastases [112,114]. Sequencing (Sanger or NGS) remains the gold standard for definitive mutation identification and is necessary in cases where IHC is negative but clinical suspicion remains high, or when rare IDH2 R172 variants (e.g., R172G, R172K, R172M) are suspected that may not be detected by available antibodies [114,115].

Immunohistochemistry is suitable for both core needle biopsy and excision specimens, offering rapid, cost-effective screening. However, a small subset of TCCRP cases (up to 10%) may be IDH2 wild-type or harbor rare R172 mutations not recognized by current antibodies, leading to false negatives [114]. Combining multiple mutation-specific antibodies can improve detection of rarer variants [115]. Sequencing is recommended for ambiguous cases or when IHC is negative but morphology is classic.

Potential pitfalls include: (1) false negatives in IDH2 wild-type TCCRP or rare R172 variants; (2) false positives are rare, as IDH2 R172 mutations are not found in other breast papillary neoplasms or in papillary thyroid carcinoma; (3) reliance solely on IHC may miss rare subtypes, so molecular confirmation is advised in challenging cases [105,112,114,115,116].

#### 3.3.5. Low-Grade Adenosquamous Carcinoma

Low-grade adenosquamous carcinoma (LGASC) of the breast is a rare variant of metaplastic breast carcinoma, accounting for fewer than 0.05% of all breast cancers. The entity was first formally described by Rosen and Ernsberger in 1987, although similar tumors were noted as early as 1912. LGASC is now recognized as a distinct clinicopathological entity due to its indolent behavior and unique histological features, which set it apart from other, more aggressive metaplastic carcinomas [27,117,118].

Histopathologically, LGASC is characterized by infiltrative small glands and nests with variable squamous differentiation, often set within a desmoplastic (fibrolamellar) stroma. Peripheral lymphocytic aggregates are common, and the tumor may mimic benign sclerosing lesions, leading to frequent misdiagnosis, especially on core needle biopsy. The presence of adenosquamous proliferation (ASP) links LGASC to a spectrum of sclerosing breast lesions, suggesting a possible biological continuum [26,27,117,118,119]. The tumor borders are typically infiltrative, and the bland cytologic appearance can mimic benign sclerosing lesions, including sclerosing adenosis and radial scar.

LGASC typically exhibits a triple-negative phenotype (ER−, PR−, HER2−), though rare cases with focal hormone receptor positivity have been reported. Myoepithelial markers such as p63 show circumferential staining around invasive glands in most cases, while smooth muscle myosin (SMM) is less consistently positive. Cytokeratin stains reveal variable patterns, and the immunoprofile can be diagnostically challenging due to inconsistent staining [118,119,120,121,122].

Imaging findings are non-specific, and LGASC often lacks distinct radiological features, further complicating preoperative diagnosis. Clinically, LGASC tends to present as a small, slow-growing mass, frequently discovered incidentally or within benign lesions such as radial scars, sclerosing lesions, or papillomas [26,27,118].

Therapeutically, wide local excision with negative margins is the mainstay of treatment. Axillary metastasis is exceedingly rare, and routine sentinel lymph node biopsy or axillary clearance is not generally indicated. The role of adjuvant radiotherapy or chemotherapy remains undefined, as most patients do not experience recurrence or metastasis following surgery alone. Over-treatment is a recognized issue due to misdiagnosis and lack of awareness of the tumor’s indolent nature [26,117,118].

Prognostically, LGASC is associated with an excellent outcome, with very low rates of local recurrence and virtually no reported distant metastases in contemporary series. However, rare cases of high-grade transformation and nodal involvement have been described, underscoring the need for complete excision and long-term follow-up. Metastatic spread is very rare, and overall survival is excellent [26,117,118,120].

The latest molecular and genetic findings indicate that low-grade adenosquamous carcinoma (LGASC) of the breast is genetically distinct from other metaplastic and high-grade triple-negative breast carcinomas. LGASC demonstrates a high frequency of PIK3CA missense mutations (found in approximately 52% of cases), most commonly in exon 20, and notably lacks TP53 mutations, which are typical of high-grade metaplastic and triple-negative breast cancers [123]. This molecular profile supports its indolent clinical behavior and basal-like phenotype.

Recent whole-genome sequencing has revealed that LGASC and its associated high-grade metaplastic carcinoma (MSC) share a GNAS c.C2530T:p.Arg844Cys mutation, which is not present in de novo MSC. Progression from LGASC to high-grade carcinoma is associated with additional pathogenic deletions in tumor suppressor genes such as KMT2D and BTG1, and loss of heterozygosity at 18q. The frequency of SMAD4::DCC fusion increases with progression, although chimeric proteins are not detected, and SMAD4 protein expression is already decreased at the LGASC stage [124].

LGASC is almost always triple-negative (ER−, PR−, HER2−), but rare cases with focal hormone receptor positivity have been reported, including in tumors with high-grade transformation [120,121]. There is also a report of LGASC occurring in a patient with a BRCA1 germline mutation, suggesting possible overlap with other basal-like breast cancers, though this association remains rare and not well characterized [125].

#### 3.3.6. Fibromatosis-like Metaplastic Carcinoma

Fibromatosis-like metaplastic carcinoma of the breast (FLMCB) is a rare, low-grade variant within the spectrum of metaplastic breast carcinomas, first systematically described by Gobbi et al. in 1999 [126]. Subsequent series, notably by Sneige et al. (2001), further characterized its clinicopathologic and immunophenotypic features [127].

Histopathologically, FLMCB is defined by a proliferation of cytologically bland spindle cells in a collagenous stroma, often with limited epithelial elements. The spindle cells infiltrate adjacent parenchyma, mimicking pure fibromatosis or scar tissue. Epithelial differentiation may be focal, with squamous or glandular elements present in some cases [126,127,128]. The tumors are typically well-circumscribed grossly but show microscopic infiltrative borders. Cytologic atypia is minimal, and mitotic activity is low, which can make diagnosis challenging.

Diagnosis relies on immunoreactivity for cytokeratins (AE1/AE3, CK5/6), which highlight epithelial differentiation, and myoepithelial markers such as p63, confirming their epithelial nature despite their deceptively mesenchymal appearance. Smooth muscle actin (SMA) may be expressed in cytokeratin-negative cells, and co-expression can occur. FLMCB is consistently negative for estrogen receptor (ER), progesterone receptor (PR), and HER2/neu, confirming its triple-negative status. Ki-67 proliferation index is low (<5%) [126,127,128,129].

Recent sequencing data indicate frequent PIK3CA H1047R and TERT promoter mutations, with no significant copy number variations, suggesting a distinct molecular pathogenesis compared to other metaplastic carcinomas [129].

FLMCB may present as an oval, high-density mass with indistinct margins on mammography and ultrasound. MRI typically shows moderate hyperintensity on T2-weighted imaging, rim enhancement, and diffusion restriction, but imaging features are not specific and may mimic benign lesions [130].

Patients with FLMCB are usually postmenopausal women (mean age ~66 years), presenting with a palpable mass. Tumor size ranges from 1–5 cm. Lymph node involvement is rare, and distant metastasis is uncommon, though isolated cases have been reported [126,127].

The mainstay of treatment for FLMCB is complete surgical excision with clear margins. Wide local excision is preferred over simple excision due to the high risk of local recurrence with inadequate margins. Axillary dissection is generally not indicated unless clinically suspicious nodes are present, given the low metastatic potential [126,127,128]. Systemic therapy is not routinely recommended due to the indolent nature and chemoresistance of these tumors [131,132].

FLMCB has a favorable prognosis among metaplastic carcinomas, with a high risk of local recurrence but low risk of lymph node or distant metastasis. Long-term survival is excellent with adequate local control [126,127,128]. The biological behavior is distinct from high-grade metaplastic carcinomas, which are more aggressive and have poorer outcomes [133].

#### 3.3.7. Mucoepidermoid Carcinoma

Mucoepidermoid carcinoma (MEC) of the breast is a rare salivary gland-type neoplasm, accounting for approximately 0.2–0.3% of all breast cancers [134,135]. The first description of breast MEC in the English-language literature dates to 1979, with subsequent case reports and series expanding the clinicopathologic understanding [134,136]. Notably, Horii et al. reported the first Japanese case in 2006 [137].

In the latest WHO Classification of Breast Tumors, MEC is defined by a mixture of mucinous, intermediate and squamoid neoplastic cells arranged in solid and cystic patterns, often with cribriform or papillary architecture [134,136,138,139]. Grading is performed using both the Auclair system (salivary gland MEC) and the Elston-Ellis system (breast carcinoma), with low-grade tumors predominating [136]. The solid component of low-grade MEC is typically composed of neoplastic cell nests with a peripheral layer of basaloid cells that gradually merge into groups of epidermoid and mucous cells. [138,140]. High-grade tumors demonstrate increased mitotic activity, nuclear atypia, and necrosis, correlating with aggressive behavior [134,136].

Immunohistochemically, MECs frequently express p63, CK5/6, CK7, and GATA3, with variable Ki-67 proliferation indices [134,135,139]. Hormone receptor expression is typically low or absent (ER: ~25%, PR: <5%, HER2: <10%), resulting in a triple-negative phenotype in 36–100% of cases [134,138,139,141]. PD-L1 and EGFR/AREG axis activation are common, and MAML2 rearrangements (CRTC1::MAML2, CRTC3::MAML2) are present in a subset, paralleling salivary gland MECs [139,140,142,143]. TP53 mutations are rare, distinguishing breast MEC from conventional high-grade triple-negative breast cancers [140].

Imaging findings are nonspecific; mammography and ultrasound often reveal a suspicious mass, sometimes with cystic components suggestive of low-grade disease [137,144]. MRI and CT may assist in preoperative staging, but definitive diagnosis relies on histopathology and immunoprofiling [144].

Clinically, MEC most commonly presents as a palpable breast mass in women with a median age of 57 years [134,138,141]. Tumors are slightly more frequent in the left breast and upper-outer quadrant [134]. Axillary lymph node involvement is uncommon in low-grade cases but portends poor prognosis when present, especially in high-grade tumors [134,145].

Therapeutic management centers on surgical resection, with modified radical mastectomy and breast-conserving surgery both utilized [134,138,139,141,145]. Lymph node dissection is performed based on clinical and pathological risk. Adjuvant therapy (chemotherapy, radiotherapy) is considered for intermediate/high-grade tumors or node-positive disease, though standardized protocols are lacking due to rarity [134,144]. Endocrine therapy is generally not indicated given the triple-negative status [139,141].

Prognosis is generally favorable for low-grade MEC, with long-term survival and low recurrence rates [134,138,139,141]. High-grade tumors, however, are associated with increased risk of nodal metastasis, distant spread, and disease-specific mortality [134,136,145]. Prognostic determinants include histologic grade, nodal status, and presence of distant metastases [134].

The overall prevalence in breast MEC is lower than in salivary gland MEC, where MAML2 fusions are present in 55–80% of cases [146,147]. However, small series and case reports have identified CRTC1::MAML2 and CRTC3::MAML2 fusions in some breast MECs, suggesting that a minority of cases may carry these rearrangements [142,148].

The clinical significance of MAML2 rearrangements in breast MEC is not fully established, but available data suggest that fusion-positive tumors are more likely to be low-grade and associated with a favorable prognosis, mirroring findings in salivary gland MEC [142,149]. Fusion-negative breast MECs may be more likely to present as high-grade tumors with aggressive behavior, although exceptions exist [145]. In salivary gland MEC, MAML2 rearrangement is an independent prognostic factor for improved overall and disease-free survival, but this has not been definitively confirmed in breast MEC due to limited case numbers [149,150].

Other actionable mutations in breast MEC are rare; TP53 mutations are infrequent, and the overall mutational burden is low [140,148]. However, enrichment of genetic alterations in the PI3K/AKT/mTOR and cell cycle regulation pathways has been observed, and EGFR/AREG axis activation is common [140]. These molecular features may have therapeutic implications: EGFR pathway activation could theoretically be targeted with anti-EGFR agents, and PI3K/AKT/mTOR pathway mutations may open avenues for targeted therapy, although clinical evidence for efficacy in breast MEC is lacking [140,151].

PD-L1 expression is frequently observed in breast MEC, suggesting a potential role for immunotherapy in selected cases [140]. However, no standard targeted therapy exists for breast MEC, and management remains primarily surgical, with adjuvant therapy considered for high-grade or advanced disease [134].

## 4. Discussion

### 4.1. Confirming Primary Origin of Triple-Negative Breast Cancer

Accurate confirmation of primary origin in low-grade triple-negative breast carcinoma (TNBC) is essential, given the diagnostic overlap with metastatic carcinomas, particularly high-grade serous carcinoma of Müllerian origin [152]. Clinically, TNBCs are more common in younger women and may present as palpable masses, often with rapid growth and aggressive features, but low-grade variants can display indolent behavior and apocrine or histiocytoid morphology [10,153,154].

Imaging clues are nuanced. TNBCs frequently lack classic suspicious features on mammography, such as spiculated margins or calcifications, and may appear as circumscribed, hypoechoic masses on ultrasound, sometimes mimicking benign lesions. MRI is the most sensitive modality for TNBC detection, reliably identifying both high- and low-grade lesions and providing superior assessment for multifocality and response to therapy. The absence of typical imaging features should prompt consideration of TNBC, especially in the context of a triple-negative immunoprofile [45,154,155,156].

Immunohistochemical markers are central to confirming breast origin. The most reliable markers for primary TNBC include TRPS1 (high sensitivity, 98.7%), SOX10 (72.8%), and GATA3 (67.5%). GCDFP15 and mammaglobin are less sensitive but highly specific for breast origin, especially in apocrine TNBCs. The androgen receptor (AR) is frequently expressed in low-grade apocrine TNBCs and supports breast origin, particularly in conjunction with GCDFP15 positivity. Basal cytokeratins (CK5/6, CK14, CK17) and EGFR are useful for identifying basal-like TNBC subtypes but are not specific for breast origin [10,153,157,158,159,160,161,162].

Pitfalls exist: PAX8 (MRQ50 clone) and WT1 may be positive in high-grade serous carcinomas but can occasionally stain TNBC, necessitating careful interpretation. Preferential use of the PAX8 BC12 clone, which is negative in TNBC, helps avoid misdiagnosis [153,163]. A panel approach, combining TRPS1, SOX10, GATA3, AR, GCDFP15, and mammaglobin, alongside exclusion of Müllerian markers (PAX8 BC12, WT1) [164], is recommended for robust confirmation of primary breast origin in triple-negative cases [10,157,161].

### 4.2. Diagnostic Algorithm and Practical Implementation

The initial recognition of LG-TNBC necessitates a systematic morphological evaluation of all triple-negative breast cancers, with a particular focus on nuclear grade, growth pattern, and specialized features. Key morphological red flags which should raise awareness include a secretory pattern with eosinophilic material, cribriform architecture with myxoid stroma, bland spindle cell proliferation, tall cells with reverse polarity, and low-grade adenosquamous features. When identifying any of those morphological red flags, the pathologist should carefully inspect the tumor for additional histopathological clues characteristic for LG-TNBC, as showcased in Table 2.

To maximize diagnostic accuracy while maintaining cost-effectiveness, a tiered immunohistochemical (IHC) approach is employed. The first-line panel includes ER, PR, and HER2 to confirm the triple-negative status, along with Ki-67 to assess the proliferation rate. A morphology-directed panel is then used, which includes Pan-TRK for secretory patterns, MYB for cribriform patterns, and IDH2 R172 for tall cell morphology. Confirmatory markers such as S100/mammaglobin for secretory patterns, SOX10/c-KIT for adenoid cystic patterns, and DOG1/lysozyme for acinic cell patterns are also utilized.

Molecular testing should be integrated for cases with positive screening markers and potential therapeutic implications. NTRK testing involves RNA-based next-generation sequencing (NGS) for ETV6-NTRK3 confirmation in Pan-TRK-positive cases. MYB testing is conducted using fluorescence in situ hybridization (FISH) or RNA in situ hybridization (RNA-ISH) for diagnostic confirmation in MYB IHC-positive cases. IDH2 sequencing is performed for mutation confirmation in IDH2 R172 IHC negative but morphologically suspicious cases.

### 4.3. Clinical Management and Treatment De-Escalation

Breast-conserving surgery represents the preferred approach for most LG-TNBCs, with wide local excision and negative margins being paramount for local control. Complete excision with clear margins reduces local recurrence risk, particularly important for entities with infiltrative growth patterns.

The 2025 management paradigm focuses on treatment de-escalation for appropriately selected patients. For chemotherapy, omission criteria include confirmed LG-TNBC subtype, T1–2 tumors, node-negative disease, clear margins, and the absence of high-risk features [49]. In terms of targeted therapy, NTRK fusion-positive tumors are eligible for TRK inhibitors, and there is potential for clinical trials for other molecular subtypes [165]. Radiation therapy is indicated following breast-conserving surgery according to standard guidelines [155].

Long-term surveillance is crucial due to the possibility of late recurrences in certain conditions. During years 1–2, patients should undergo a clinical examination every 3–4 months, with imaging conducted as necessary. From years 3–5, clinical examinations should be performed every 6 months, accompanied by annual mammography. Beyond 5 years, annual clinical and imaging surveillance is recommended, with an emphasis on being aware of the potential for late recurrence.

The omission of chemotherapy and axillary surgery in low-grade triple-negative breast cancer (LG-TNBC) is supported for several special histologic subtypes and for small, node-negative tumors, based on consensus statements, retrospective series, and guideline recommendations. The recurrence risk for these subtypes is generally low, especially for pure forms and pT1aN0 TNBC, and management can often be de-escalated to surgery alone. However, prospective, long-term outcome data and randomized trials are lacking for these rare entities. The table below (Table 3) summarizes recurrence risk and management recommendations for each LG-TNBC subtype to enhance clinical relevance.

### 4.4. Sentinel Lymph Node Biopsy in Low-Grade Triple-Negative Breast Carcinoma

The role of sentinel lymph node biopsy (SLNB) in low-grade triple-negative breast carcinoma is recommended only in certain scenarios, reflecting a nuanced approach based on tumor biology, response to therapy, and patient characteristics. The American Society of Clinical Oncology (ASCO) recommends SLNB as the standard axillary staging procedure for clinically node-negative (cN0) invasive breast cancer, including triple-negative subtypes, except in select low-risk populations [167].

In the neoadjuvant setting, recent pooled analyses indicate that patients with triple-negative breast cancer (TNBC) who are cN0 at diagnosis and achieve a pathologic complete response (pCR) after neoadjuvant systemic therapy (NAST) have a very low rate of sentinel node positivity (ypN+ rate ~2.2%), suggesting that omission of SLNB may be considered in this highly selected group [168]. Nevertheless, this approach remains investigational and is not yet standard of care, as ongoing trials are needed to confirm long-term outcomes and safety [168,169]. For TNBC patients who do not achieve pCR or who present with clinically node-positive disease, SLNB remains indicated for axillary staging, with axillary lymph node dissection (ALND) reserved for those with residual nodal disease [167,169,170].

Technical considerations are critical, as false-negative rates for SLNB after neoadjuvant therapy are higher, particularly in patients who convert from cN+ to cN0 status; thus, careful multidisciplinary assessment and localization of previously positive nodes are essential [170]. Pathologic evaluation should focus on detection of macrometastases, with cytokeratin immunohistochemistry reserved for post-neoadjuvant cases where residual disease is suspected [169,171].

In summary, SLNB is recommended for most patients with low-grade triple-negative breast carcinoma who are clinically node-negative, except for those with exceptional response to neoadjuvant therapy, where omission may be considered but is not yet standard practice. Routine omission of SLNB is not advised outside of clinical trials for TNBC, given its aggressive biology and higher risk of nodal involvement [167,168,172].

### 4.5. Immunotherapy in Low-Grade Triple-Negative Breast Cancer

Immunotherapy has emerged as a significant therapeutic option in triple-negative breast carcinoma (TNBC), including low-grade variants, due to the immunogenic nature of this subtype and the presence of tumor-infiltrating lymphocytes and PD-L1 expression [173,174]. The clinical utility of immune checkpoint inhibitors (ICIs), such as pembrolizumab and atezolizumab, is now well established in metastatic TNBC, contingent upon PD-L1 status.

PD-L1 testing is recommended only in specific scenarios. According to the American Society of Clinical Oncology, PD-L1 testing should be performed in patients with locally recurrent unresectable or metastatic TNBC who are candidates for ICI-based therapy, specifically to determine eligibility for pembrolizumab plus chemotherapy [84].

In contrast, PD-L1 testing is not currently recommended in early-stage TNBC to guide immunotherapy decisions, as randomized studies have shown that the benefit of neoadjuvant immunotherapy appears independent of PD-L1 status, and there is insufficient evidence to support routine PD-L1 testing in this setting [175].

Meta-analyses and clinical trials confirm that survival benefits from ICIs are most pronounced in PD-L1-positive metastatic TNBC, but immunotherapy can also increase the risk of immune-related adverse events, necessitating careful patient selection and risk assessment [176]. Furthermore, genomic and transcriptomic heterogeneity may affect the predictive value of PD-L1 and tumor-infiltrating lymphocytes, underscoring the need for ongoing research into more precise biomarkers [177].

However, the efficacy of immune checkpoint inhibitors in LG-TNBCs specifically remains uncharacterized, as clinical trials have not reported outcomes for this subgroup [174,178,179,180,181,182].

Current evidence suggests that PD-L1 is an imperfect biomarker, with spatial and temporal heterogeneity and lack of consensus on testing methodology. Novel biomarkers such as tumor mutational burden (TMB), microsatellite instability/mismatch repair deficiency (MSI/dMMR), and immune gene signatures are under investigation to better predict response to immunotherapy. Multi-omics approaches and molecular subtyping, as exemplified by the FUTURE-SUPER trial, may facilitate precision treatment tailored to specific TNBC subtypes, including immunomodulatory variants that may be more responsive to immune-based therapies [179,183,184,185].

Future research should prioritize dedicated studies of LG-TNBCs, integrating comprehensive molecular and immunologic profiling to clarify their immunogenicity and therapeutic vulnerabilities. Combination strategies—such as pairing immune checkpoint inhibitors with targeted agents, antibody-drug conjugates, or cellular therapies—represent promising avenues to enhance clinical benefit in these rare subtypes.

### 4.6. Low-Grade Triple-Negative Breast Carcinoma, Not Otherwise Specified

Interpretation of triple-negative breast carcinomas (TNBCs) with low-grade nuclear features, low mitotic index, and low Ki-67 proliferation rate, but without the classic morphologic characteristics of recognized low-grade TNBC subtypes, requires careful clinicopathologic correlation. These tumors, though rare, constitute a biologically and clinically distinct subgroup within TNBC, often demonstrating indolent behavior and favorable prognosis compared to conventional high-grade TNBCs [8,9,10,49,61,186].

Classification should be based on integration of histologic grade, proliferation indices (Ki-67 ≤30%), and molecular features. While most low-grade TNBCs are associated with specific morphologies (e.g., apocrine, secretory, adenoid cystic), cases lacking these features but retaining low-grade cytology and low proliferation should be recognized as part of the broader spectrum of low-risk TNBCs [8,9,10,49,61,186]. Molecular profiling may reveal enrichment for PI3K/AKT pathway alterations and androgen receptor expression, but these are not universal [10,186]. The operational term “triple-negative” encompasses significant heterogeneity, and recognition of these low-grade variants is essential for precision medicine approaches [8,9,61]. In the literature, these cases are most commonly reported as “low-grade triple-negative breast carcinoma, not otherwise specified” (NOS), or as “low-grade triple-negative breast carcinoma” with an explicit comment indicating that they do not fit into any established category, such as secretory, adenoid cystic, or apocrine carcinoma [8,9,61,187].

Management of these tumors should be individualized. Evidence indicates that systemic chemotherapy offers limited benefit in stage I low-grade TNBCs with low Ki-67, and de-escalation of therapy may be appropriate in select cases [49,61,186,188]. Tumor stage remains the most important prognostic factor; for stage I disease, observation or surgery alone may suffice, while stage II or higher may warrant consideration of adjuvant therapy, though the benefit is less clear in low-proliferation tumors [186,188]. The prognostic value of Ki-67 is particularly relevant, as patients with Ki-67 ≤30% and early-stage disease have excellent outcomes, and adjuvant chemotherapy does not significantly improve disease-free or overall survival in this group [186,188]. Multidisciplinary evaluation is recommended for optimal management [49].

### 4.7. BRCA Mutations in Low-Grade Triple-Negative Breast Cancer

BRCA gene mutations, particularly in BRCA1, are significantly enriched in patients with triple-negative breast cancer (TNBC) compared to other breast cancer subtypes. Large cohort studies of TNBC, unselected for family history, report that deleterious BRCA1/2 mutations are present in approximately 10–15% of TNBC cases, with BRCA1 mutations being more frequent than BRCA2 [189,190,191,192,193]. This prevalence is notably higher in younger patients and those with a family history of breast or ovarian cancer [191,193,194].

When low-grade TNBCs do occur, the frequency of BRCA mutations appears lower than in high-grade TNBCs. Nearly all BRCA1 mutation carriers with TNBC present with grade III tumors, and low-grade TNBCs are infrequently associated with BRCA1 mutations [193].

Beyond BRCA1/2, other homologous recombination repair genes (e.g., PALB2, BARD1, RAD51D) are also implicated in TNBC, but their association with low-grade histology is not well established [189,195]. The concept of “BRCAness”—tumors with molecular features similar to BRCA-mutated cancers—further complicates the landscape, as many sporadic TNBCs exhibit BRCA1-like genomic profiles, but these are also predominantly high-grade [196,197].

In summary, low-grade triple-negative breast cancers tend to appear more rarely in patients with BRCA gene mutations, and when BRCA mutations are present in TNBC, they are most commonly associated with high-grade, basal-like, or unclassified molecular subtypes, rather than low-grade or LAR subtypes [189,196,198].

### 4.8. Evidence Quality and Strength of Recommendations

The strength of evidence supporting clinical recommendations for LG-TNBCs varies considerably across entities, reflecting differences in disease rarity, historical recognition, and availability of molecular characterization techniques. Understanding this evidence hierarchy is essential for appropriate interpretation of the literature and clinical decision-making.

Among the seven recognized LG-TNBC entities, secretory carcinoma demonstrates the highest-quality evidence base. This entity benefits from multiple large institutional series, including a recent multicenter study of 80 molecularly confirmed cases [35], definitive identification of the pathognomonic ETV6-NTRK3 fusion, and most importantly, high-quality prospective therapeutic data from Phase I/II trials demonstrating an objective response rate of 75% for TRK inhibitors in advanced disease [82,83,84]. These trials led to FDA and EMA approval for tissue-agnostic use of larotrectinib and entrectinib in NTRK fusion-positive solid tumors, establishing secretory carcinoma as the only LG-TNBC entity with Level 1 evidence for targeted systemic therapy.

Adenoid cystic carcinoma and tall cell carcinoma with reversed polarity represent entities with moderate to high-quality evidence. For adenoid cystic carcinoma, multiple retrospective institutional series comprising aggregate numbers exceeding 200 cases, combined with well-characterized MYB/MYBL1 rearrangements and long-term outcome data, provide a robust evidence base [63,64,65,66,67,68,69,70,71,72,73]. The recent recognition of the solid-basaloid variant with distinct biology and worse prognosis further exemplifies the evolution of evidence in this entity [63,64,65,66,67,68]. TCCRP, though more recently characterized, benefits from highly specific molecular validation through IDH2 R172 mutation testing with 75–85% sensitivity, excellent reproducibility of morphologic diagnosis, and remarkably favorable outcomes (recurrence rate 2.2%, OS approaching 100%) documented across multiple series [105,106,107,108,109,110,111,112,113,114,115,116].

Acinic cell carcinoma, low-grade adenosquamous carcinoma, and fibromatosis-like metaplastic carcinoma are supported by moderate-quality evidence consisting primarily of retrospective case series and molecular characterization studies. For acinic cell carcinoma, approximately 60–100 cases have been reported with molecular studies definitively distinguishing breast primaries from salivary gland counterparts through absence of NR4A3 rearrangements [89,98,99,100,101,102,103,104]. However, the heterogeneous terminology and reported adverse outcomes in over 25% of cases necessitate cautious interpretation [21,96,97]. Similarly, LGASC and FLMCB are characterized predominantly through case series with molecular validation of PIK3CA mutations and TERT promoter alterations, but limited aggregate patient numbers restrict definitive prognostic statements [123,124,129].

Mucoepidermoid carcinoma represents the entity with the lowest-quality evidence, with fewer than 30 well-documented cases in the breast literature and reliance primarily on single case reports [134,135]. While morphologic and immunohistochemical criteria are well-established through analogy with salivary gland counterparts, the inconsistent identification of MAML2 rearrangements and limited long-term outcome data preclude strong evidence-based recommendations.

The implications of this evidence hierarchy for clinical practice are significant. For secretory carcinoma, the high-quality evidence supports definitive recommendations for molecular testing and consideration of targeted therapy in advanced disease. For entities with moderate evidence quality, recommendations must be more cautious, emphasizing the need for complete surgical excision while acknowledging uncertainty regarding optimal adjuvant approaches. For mucoepidermoid carcinoma, management recommendations are necessarily extrapolated from related entities and expert consensus.

Table 4 summarizes the evidence quality assessment for each LG-TNBC entity, incorporating study design, sample sizes, molecular validation quality, outcome data availability, and therapeutic evidence.

### 4.9. Future Directions

The rarity of LG-TNBCs poses challenges in developing evidence-based treatment guidelines, as most data derive from case reports and small series. Collaborative registries and molecularly annotated datasets are needed to better characterize their clinical trajectory, refine diagnostic criteria, and identify novel therapeutic vulnerabilities. Furthermore, the question of whether all LG-TNBCs represent a biologically unified group or simply a heterogeneous collection of salivary gland-like tumors occurring in the breast remains open. Molecular profiling studies suggest both shared features (e.g., indolent clinical behavior, low TP53 mutation frequency) and unique, entity-specific drivers.

Despite recent advances in the molecular characterization of low-grade triple-negative breast cancers (LG-TNBCs), significant gaps remain in the understanding of their long-term clinical outcomes and optimal therapeutic strategies. There is a paucity of prospective, subtype-specific data, particularly regarding the efficacy and durability of targeted therapies such as tropomyosin receptor kinase (TRK) and phosphoinositide 3-kinase (PI3K) pathway inhibitors. While multi-omics profiling has identified actionable kinase-driven alterations and PI3K/Akt/mTOR pathway dysregulation in subsets of TNBC, clinical trials of these agents have yielded limited efficacy in unselected populations, and resistance mechanisms remain a major barrier to durable responses [199,200,201,202].

The absence of robust long-term outcome data for LG-TNBCs, including recurrence rates, survival, and quality-of-life metrics, impedes the development of evidence-based guidelines and risk-adapted management. Priority areas for collaborative research include the establishment of international registries and molecularly annotated cohorts to enable longitudinal outcome tracking, the design of prospective clinical trials stratified by molecular subtype, and the integration of biomarker-driven approaches to optimize patient selection for targeted therapies [179,203,204]. In particular, studies should focus on evaluating the clinical utility of TRK and PI3K pathway inhibitors in LG-TNBCs, elucidating resistance mechanisms, and exploring rational combination strategies with immunotherapy or other targeted agents.

### 4.10. Limitations and Challenges

Current limitations include the rarity of individual entities limiting robust clinical trial design, variability in diagnostic practices across institutions, and limited long-term follow-up data for newer molecular subtypes. Healthcare access disparities may limit molecular testing availability, particularly in resource-constrained settings.

Future research priorities include prospective validation of de-escalation strategies, development of predictive biomarkers for treatment selection, and investigation of resistance mechanisms to targeted therapies. International collaborative registries are essential for accumulating sufficient data on these rare entities.

## 5. Conclusions

Low-grade triple-negative breast carcinomas (LG-TNBCs) constitute a rare but clinically important subset of breast cancers. Despite their triple-negative immunoprofile, these tumors differ markedly from conventional high-grade TNBC in morphology, molecular alterations, clinical behavior, and prognosis. Recognition of distinct entities—including adenoid cystic carcinoma, secretory carcinoma, acinic cell carcinoma, tall cell carcinoma with reversed polarity, low-grade adenosquamous carcinoma, fibromatosis-like metaplastic carcinoma, and mucoepidermoid carcinoma—is essential to avoid overtreatment and to guide patient management appropriately.

Molecular characterization plays a growing role in diagnosis and precision therapy, exemplified by the identification of ETV6–NTRK3 fusions in secretory carcinoma and MYB rearrangements in adenoid cystic carcinoma. Surgical excision with clear margins remains the mainstay of treatment for most LG-TNBCs, while systemic therapy is generally reserved for rare high-grade or metastatic cases.

Future studies leveraging multicenter registries and molecular profiling are needed to refine diagnostic criteria, clarify natural history, and identify additional therapeutic targets. Ultimately, heightened awareness and careful characterization of LG-TNBCs allow clinicians and pathologists to provide prognostically accurate counseling and personalized management, improving outcomes while minimizing unnecessary interventions.

## Figures and Tables

**Table 1 cancers-17-03635-t001:** Summary of Low-Grade Triple-Negative Breast Carcinomas.

Entity	Median Age at Diagnosis	Prevalence Among TNBCs	Typical Histology	Immunoprofile	Molecular Alterations	Prognosis
Adenoid cystic carcinoma	~62 years [48]	~0.1–0.7% [9,49,50,51]	Cribriform/tubular/solid with basement membrane-like material; dual cell population	ER−/PR−/HER2−; CK5/6+, p63+, c-KIT+	MYB–NFIB or MYBL1 rearrangements	Excellent prognosis, rare metastasis; surgery usually sufficient
Secretory carcinoma	Wide age range; ~25–40 years; can occur in children and adults [52,53]	~0.1–0.2% [9,49,54]	Microcystic/tubular/solid, abundant eosinophilic secretions	ER−/PR−/HER2−; S100+, mammaglobin+	ETV6–NTRK3 fusion (pathognomonic)	Excellent prognosis; TRK inhibitors effective in advanced disease
Acinic cell carcinoma	~50–60 years [53,55]	~0.1% [9,49,52]	Resembles salivary acinic cell carcinoma; granular cytoplasm	ER−/PR−/HER2−; DOG1+, lysozyme+	No recurrent driver identified; complex karyotype	Indolent, but local recurrence possible
Tall cell carcinoma with reversed polarity	~65 years [53]	~0.1% [9,49,55]	Tall cells with apical snouts, reverse polarity nuclei	ER−/PR−/HER2−; CK5/6+, TTF1−	IDH2 mutations, PIK3CA co-mutations	Generally favorable; may mimic papillary lesions
Low-grade adenosquamous carcinoma	~60 years [53]	~0.1% [9,49]	Infiltrative small glands, squamous differentiation, desmoplastic stroma	ER−/PR−/HER2−; CK5/6+, p63+	PIK3CA mutations in subset	Indolent, local recurrence possible
Fibromatosis-like metaplastic carcinoma	~50–60 years [53]	~0.1% [9,49]	Bland spindle cells, mimics fibromatosis	ER−/PR−/HER2−; CK+, p63+	No specific recurrent alteration	Locally aggressive, rare metastasis
Mucoepidermoid carcinoma	~50–60 years [53]	~0.1% [9,49,53]	Mixture of mucinous, squamoid, and intermediate cells	ER−/PR−/HER2−; CK5/6+, mucicarmine+	CRTC1–MAML2 fusion (in subset)	Rare, usually indolent but limited data

**Table 2 cancers-17-03635-t002:** Diagnostic algorithm for accurate identification and classification of LG-TNBC.

Morphological ‘Red Flags’	Histopathological Features	IHC	Molecular
Secretory pattern with eosinophilic material	Microcystic/tubular growth, abundant pink secretions, bland cytology	Pan-TRK, S100, mammaglobin	RNA-based NGS for ETV6-NTRK3 (if Pan-TRK+)
Cribriform architecture	Dual cell population, basement membrane material, myxoid stroma	MYB, SOX10, c-KIT, CK5/6	MYB FISH or RNA-ISH (if MYB 3+ diffuse)
Tall cells with reverse polarity	Apical nuclei, nuclear grooves, pseudoinclusions, papillary pattern	IDH2 R172 (11C8B1), GATA3, CK7	Sequencing if IDH2 IHC negative
Granular eosinophilic cytoplasm	Acinic cell differentiation, zymogen granules, solid/microglandular	DOG1, lysozyme, NR4A3	TP53/PIK3CA sequencing
Low-grade adenosquamous features	Small glands, squamous differentiation, desmoplastic stroma	CK5/6, p63, p40	PIK3CA sequencing
Bland spindle cell proliferation	Fibromatosis-like appearance, minimal atypia, low Ki-67	Pan-CK, p63, SMA	PIK3CA/TERT sequencing
Mixed cell populations	Mucinous/squamoid/intermediate cells, cystic spaces	Mucicarmine, CK5/6, GATA3	MAML2 FISH (selected cases)

**Table 3 cancers-17-03635-t003:** Recurrence risk and management recommendations for LG-TNBC.

Histologic Subtype	Evidence for Omission of Chemotherapy	Evidence for Omission of Axillary Surgery	Recurrence Risk	Management Recommendations	Key References
Adenoid cystic carcinoma (classical)	Supported; favorable prognosis, consensus recommends omission in pure forms	Supported; omission reasonable in small, node-negative cases	Low; indolent, rare recurrence	Surgery alone; omit chemotherapy and axillary surgery if node-negative	[49,61]
Low-grade adenosquamous carcinoma	Supported; consensus and retrospective data support omission	Supported; omission reasonable in small, node-negative cases	Low; indolent, rare recurrence	Surgery alone; omit chemotherapy and axillary surgery if node-negative	[49,61]
Fibromatosis-like metaplastic carcinoma	Supported; consensus and retrospective data support omission	Supported; omission reasonable in small, node-negative cases	Low; indolent, rare recurrence	Surgery alone; omit chemotherapy and axillary surgery if node-negative	[49,61]
Low-grade mucoepidermoid carcinoma	Supported; consensus and retrospective data support omission	Supported; omission reasonable in small, node-negative cases	Low; indolent, rare recurrence	Surgery alone; omit chemotherapy and axillary surgery if node-negative	[49,61]
Secretory carcinoma	Supported; consensus and retrospective data support omission	Supported; omission reasonable in small, node-negative cases	Low; indolent, rare recurrence	Surgery alone; omit chemotherapy and axillary surgery if node-negative	[49,61]
Acinic cell carcinoma	Supported; consensus and retrospective data support omission	Supported; omission reasonable in small, node-negative cases	Low; indolent, rare recurrence	Surgery alone; omit chemotherapy and axillary surgery if node-negative	[49,61]
Tall cell carcinoma with reversed polarity	Supported; consensus and retrospective data support omission	Supported; omission reasonable in small, node-negative cases	Low; indolent, rare recurrence	Surgery alone; omit chemotherapy and axillary surgery if node-negative	[49,61]
Small, node-negative TNBC (pT1aN0)	Supported; guidelines and outcome data support omission	Supported; omission reasonable in small, node-negative cases	Low; 5-year RFS > 90%	Surgery alone; omit chemotherapy and axillary surgery	[58,61,62]
Small, node-negative TNBC (pT1bN0)	Case-by-case; benefit of chemotherapy is modest, omission may be considered for low-risk features	Supported; omission reasonable in small, node-negative cases	Low-moderate; recurrence risk higher than pT1a	Surgery alone or consider chemotherapy for high-risk features; omit axillary surgery	[58,62]
Small, node-negative TNBC (pT1cN0)	Not supported; chemotherapy recommended, improves recurrence-free survival	Supported; omission reasonable in small, node-negative cases	Moderate; recurrence risk higher than pT1a/b	Surgery plus chemotherapy; omit axillary surgery if node-negative	[58,62,166]

**Table 4 cancers-17-03635-t004:** Strength of Evidence for Each LG-TNBC Entity.

Entity	Evidence Level	Justification
Adenoid Cystic Carcinoma	Moderate to High	Multiple retrospective series (aggregate >200 cases), well-characterized MYB/MYBL1 rearrangements, long-term outcome data available, distinction between classic and solid-basaloid variants established
Secretory Carcinoma	High	Large multicenter series (*n* = 80 with molecular confirmation), pathognomonic ETV6-NTRK3 fusion, 5-year BCSS >95%, Phase I/II trials of TRK inhibitors with FDA/EMA approval
Acinic Cell Carcinoma	Moderate	Multiple case series (aggregate 60–100 cases), molecular distinction from salivary gland counterpart (NR4A3 absent), but heterogeneous terminology and >25% adverse outcomes
Tall Cell Carcinoma with Reversed Polarity	Moderate	Multiple case series (>50 cases), highly specific IDH2 R172 mutation (75–85%), excellent outcomes (recurrence 2.2%, OS ~100%), but relatively recently characterized
Low-Grade Adenosquamous Carcinoma	Moderate	Limited case series (<50 total), PIK3CA and GNAS mutations identified, excellent outcomes but risk of high-grade transformation documented
Fibromatosis-like Metaplastic Carcinoma	Moderate	Multiple case series (aggregate 50–100), PIK3CA H1047R and TERT mutations, favorable prognosis but high local recurrence risk
Mucoepidermoid Carcinoma	Moderate to Low	Very limited data (<30 cases), predominantly case reports, MAML2 rearrangements in subset only, insufficient long-term data

## Data Availability

No new data were created or analyzed in this study. Data sharing is not applicable to this article.

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
