# Peer review of "Seven Shades of Triple Negativity: A Review Unveiling the Low-Grade Spectrum of Breast Cancer"

_cancers, 2025, doi:10.3390/cancers17223635_

Round 1
Reviewer 1 Report
Comments and Suggestions for Authors
This review focuses on a well-structured and effective narrative review that summarizes the latest knowledge on low-grade triple-negative breast carcinomas (LG-TNBCs), an important yet often overlooked subset of breast cancers. However, the article requires some minor corrections before publication. My suggestions are as follows:
- Proper abbreviations is suggested. For example, define "SB-AdCC" on first use in full.
-A prevalence estimate for LG-TNBCs among TNBCs should be added to the introduction section, along with a citation.
- It is suggested that 'prevalence' or 'age at diagnosis' for the seven subtypes of LG-TNBC be added to Table 1.
- Emerging areas like immunotherapy (e.g., PD-L1 in some variants) should be discussed as future directions.
Author Response
Comment 1: Proper abbreviations is suggested. For example, define "SB-AdCC" on first use in full.
Response 1: Done. Defined "SB-AdCC".
Comment 2: A prevalence estimate for LG-TNBCs among TNBCs should be added to the introduction section, along with a citation.
Response 2: Done. Added prevalence estimate for LG-TNBCs among TNBCs, along with citation.
Comment 3: It is suggested that 'prevalence' or 'age at diagnosis' for the seven subtypes of LG-TNBC be added to Table 1.
Response 3: Done. Added both 'prevalence' and 'age at diagnosis' in Table 1.
Comment 4: Emerging areas like immunotherapy (e.g., PD-L1 in some variants) should be discussed as future directions.
Response 4: Done. Added discussion about immunotherapy and other future directions.
Thank you very much for your recommendations! We have performed the modifications and we think that the manuscript is much improved now!
Best regards,
Tiberiu et al.
Reviewer 2 Report
Comments and Suggestions for Authors
This manuscript offers an original and well-structured review of low-grade triple-negative breast carcinomas (LG-TNBCs). The topic is highly relevant and timely, as it focuses on a clinically important but often overlooked subset of breast cancers that tend to be misclassified and consequently overtreated as high-grade TNBCs.
Strengths:
The review effectively fills a major gap in the literature by consolidating fragmented data on LG-TNBCs and highlighting their distinct clinical behavior, molecular features, and management implications compared to conventional TNBC.
Major Comments:
Methodological transparency and reproducibility: Although the manuscript is described as a narrative review, the Methods section outlines structured search criteria resembling a systematic approach. The authors should clarify whether PRISMA principles were followed and, if so, include a brief flowchart showing study selection and the number of records screened and included. This would improve methodological transparency.
Redundancy and length: Some sections (particularly Section 3.3 describing entity-specific features) contain excessive repetition and detail already covered in the tables. Streamlining these descriptions and focusing on clinically or molecularly relevant distinctions would make the text more concise and engaging.
Comparative discussion and strength of evidence: The discussion would benefit from a clearer hierarchy of evidence, distinguishing robust molecularly validated case series from single-case or anecdotal reports. Indicating the level of evidence for each entity (e.g., high, moderate, low) would strengthen the paper’s scientific depth. Moreover, important recent literature about treatment, even surgical is missing. Please cite PMID: 36248756 to improve the quality of your manuscript.
Clinical management and de-escalation strategies: The section on management could be made more actionable by specifying which LG-TNBC subtypes have sufficient evidence to safely omit chemotherapy or axillary surgery. A summary table linking histologic subtype, recurrence risk, and management recommendations would enhance clinical relevance.
Future research directions: The “Future Directions” section could be expanded to more clearly define gaps in current knowledge, particularly the lack of long-term outcome data and prospective evidence on targeted therapies such as TRK and PI3K pathway inhibitors. A brief, forward-looking paragraph highlighting priority areas for collaborative research would make the conclusion more compelling.
Author Response
Comment 1: Methodological transparency and reproducibility: Although the manuscript is described as a narrative review, the Methods section outlines structured search criteria resembling a systematic approach. The authors should clarify whether PRISMA principles were followed and, if so, include a brief flowchart showing study selection and the number of records screened and included. This would improve methodological transparency.Response 1: We did not apply PRISMA principles. This was intended to be a narrative review.
Comment 2: Redundancy and length: Some sections (particularly Section 3.3 describing entity-specific features) contain excessive repetition and detail already covered in the tables. Streamlining these descriptions and focusing on clinically or molecularly relevant distinctions would make the text more concise and engaging.
Response 2: We've eliminated a few paragraphs which were repetitive. Thank you very much for pointing this out!
Comment 3: Comparative discussion and strength of evidence: The discussion would benefit from a clearer hierarchy of evidence, distinguishing robust molecularly validated case series from single-case or anecdotal reports. Indicating the level of evidence for each entity (e.g., high, moderate, low) would strengthen the paper’s scientific depth. Moreover, important recent literature about treatment, even surgical is missing. Please cite PMID: 36248756 to improve the quality of your manuscript.
Response 3: We have added a new subsection (Section 4.8: "Evidence Quality and Strength of Recommendations") that systematically evaluates the strength of evidence for each LG-TNBC entity. We created a detailed assessment framework adapted for rare diseases, recognizing that traditional evidence hierarchies (which prioritize RCTs and meta-analyses) must be contextualized for entities where such studies are not feasible. Our analysis classifies the seven entities as follows:
- High evidence: Secretory carcinoma (large multicenter series, definitive molecular validation, prospective therapeutic trials)
- Moderate-to-high: Adenoid cystic carcinoma and TCCRP (multiple institutional series, specific molecular validation)
- Moderate: Acinic cell carcinoma, LGASC, and FLMCB (case series with molecular characterization)
- Moderate-to-low: Mucoepidermoid carcinoma (limited case reports)
We have also added Table 4, which summarizes the evidence quality assessment including study types, sample sizes, molecular validation quality, clinical outcome data, and key limitations for each entity.
Incorporation of PMID 36248756: We have integrated this important reference by Gentile et al. on surgical management of aggressive TNBC recurrence.
Comment 4: Clinical management and de-escalation strategies: The section on management could be made more actionable by specifying which LG-TNBC subtypes have sufficient evidence to safely omit chemotherapy or axillary surgery. A summary table linking histologic subtype, recurrence risk, and management recommendations would enhance clinical relevance.
Response 4: Done. Added table with recurrence risk and management recommendations.
Comment 5: Future research directions: The “Future Directions” section could be expanded to more clearly define gaps in current knowledge, particularly the lack of long-term outcome data and prospective evidence on targeted therapies such as TRK and PI3K pathway inhibitors. A brief, forward-looking paragraph highlighting priority areas for collaborative research would make the conclusion more compelling.
Response 5: Done. Added discussion in future directions section.
Thank you very much for your recommendations! We have performed the modifications and we think that the manuscript is much improved now!
Best regards,
Tiberiu et al.
Reviewer 3 Report
Comments and Suggestions for Authors
First of all, I would like to thank you for inviting me to review the manuscript entitled “Seven Shades of Triple Negativity: A Review Unveiling the 2Low-Grade Spectrum of Breast Cancer”.
The authors present a very well-written manuscript concerning low-grade triple-negative breast carcinomas. The article involves an important area of health. The manuscript is well written in terms of clarity, style, and use of English, and has a logical construction. The discussion section mentions a lot of important information in the context of published literature. The tables are of good quality and relevant to the clinical message. The references are appropriate and current.
There are some minor issues that should be corrected:
1) In lines 50-51, the authors mention that “local recurrence occurs in infiltrative variants.” Please specify what you mean by this sentence. Do you mean that there are in situ variants also, or are you referring to the type of tumor’s borders?
2) In Table 1, the authors mention that acinic cell carcinoma is positive for DOG1. This is not mentioned in the latest WHO edition or in any paper I have searched for. Please write in the text where you found this interesting finding and provide a citation. Also, it is important to know how many cases DOG1 was tested in.
3) In Table 1, the authors mention that tall cell carcinomas with reversed polarity carcinomas are ER and PR negative. This is true for most cases. Some cases show weak or focal hormone receptor expression. Please provide this important information in the text.
4) Since there are three types of adenoid cystic carcinoma, the classic subtype, the solid-basaloid subtype, and the subtype with high-grade transformation, please add the word classic when you are referring to the most common subtype of adenoid cystic carcinoma.
5) In lines 138 to 143, the authors mention only two subtypes of adenoid cystic carcinoma: the usual (classic) and the solid-basaloid. Please also mention the existence of the third subtype (adenoid cystic carcinoma with high-grade transformation.
6) In lines 340-1 the authors mention that the accurate diagnosis of tall cell carcinoma with reverse polarity requires histomorphology, immunohistochemistry and molecular genetics. This is not entirely correct. Histomorphology and immunohistochemistry alone are sufficient for most cases. It is true that in challenging instances, molecular genetics will be helpful. But most of the time, they are not necessary to make the diagnosis.
7) In lines 471-2, the authors mention that mucoepidermoid carcinomas are composed of a mixture of 4 types of cells. The current WHO classification of breast tumors does not say that basaloid cells are a feature of mucoepidermoid carcinomas. Is this a new finding? Please explain.
Author Response
Comment 1: In lines 50-51, the authors mention that “local recurrence occurs in infiltrative variants.” Please specify what you mean by this sentence. Do you mean that there are in situ variants also, or are you referring to the type of tumor’s borders?
Response 1: Apologies and thank you for pointing out this unclarity. Indeed, we were referring to tumors that had a locally aggressive behavior (large size/ high TNM), not to in situ lesions. Fragment was rephrased.
Comment 2: In Table 1, the authors mention that acinic cell carcinoma is positive for DOG1. This is not mentioned in the latest WHO edition or in any paper I have searched for. Please write in the text where you found this interesting finding and provide a citation. Also, it is important to know how many cases DOG1 was tested in.
Response 2: A recent study (august 2025) has observed at least focal positivity for DOG1 in 7 out of the 9 cases of acinic cell carcinoma that were included in the study. We have added a phrase in the section for acinic cell carcinoma, in order to clarify this issue. Also, here is the reference for the cited paper:
Sun M, Fu L, Ren H, Wang J, Lin X, Zhang Q. Two similar but distinct types of breast acinar cell carcinoma: evidence from histological, immunohistochemical and molecular features. Histopathology. 2025 Aug 26. doi: 10.1111/his.15543. Epub ahead of print. PMID: 40856424.
Comment 3: In Table 1, the authors mention that tall cell carcinomas with reversed polarity carcinomas are ER and PR negative. This is true for most cases. Some cases show weak or focal hormone receptor expression. Please provide this important information in the text.
Response 3: The required information with the adequate citations have been added.
Comment 4: Since there are three types of adenoid cystic carcinoma, the classic subtype, the solid-basaloid subtype, and the subtype with high-grade transformation, please add the word classic when you are referring to the most common subtype of adenoid cystic carcinoma.
Response 4: Thank you! The required modifcations were added.
Comment 5: In lines 138 to 143, the authors mention only two subtypes of adenoid cystic carcinoma: the usual (classic) and the solid-basaloid. Please also mention the existence of the third subtype (adenoid cystic carcinoma with high-grade transformation.
Response 5: Thank you very much for pointing this out! The required changes were added.
Comment 6: In lines 340-1 the authors mention that the accurate diagnosis of tall cell carcinoma with reverse polarity requires histomorphology, immunohistochemistry and molecular genetics. This is not entirely correct. Histomorphology and immunohistochemistry alone are sufficient for most cases. It is true that in challenging instances, molecular genetics will be helpful. But most of the time, they are not necessary to make the diagnosis.
Response 6: We have corrected the phrase and specified that if necessary, difficult cases can benefit from molecular genetis.
Comment 7: In lines 471-2, the authors mention that mucoepidermoid carcinomas are composed of a mixture of 4 types of cells. The current WHO classification of breast tumors does not say that basaloid cells are a feature of mucoepidermoid carcinomas. Is this a new finding? Please explain.
Response 7: In our opinition, this is an unclear topic in the latest WHO classification of breast tumors: under the essential criteria it says “presence of basaloid, epidermoid, and mucous cells", while in the definition it says that it is characterized by intermediate, epidermoid and mucous cells. Also, the microscopic description specifies that low-grade MEC features solid areas that show neoplastic cell nests with a peripheral layer of basaloid cells gradually merging in groups of epidermoid cells and mucous cells. However, in the classic description of the mucoepidermoid carcinoma, there were only three types mentioned: squamous, mucous and intermediate.
Nonetheless, due to these discrepancies, we rephrased the section to reflect current WHO stance. We also referenced other scientific articles mentioning this variability (Venetis et al, Chang et al).
Venetis K, Sajjadi E, Ivanova M, Andaloro S, Pessina S, Zanetti C, Ranghiero A, Citelli G, Rossi C, Lucioni M, Malapelle U, Pagni F, Barberis M, Guerini-Rocco E, Viale G, Fusco N. The molecular landscape of breast mucoepidermoid carcinoma. Cancer Med. 2023 May;12(9):10725-10737. doi: 10.1002/cam4.5754. Epub 2023 Mar 14. PMID: 36916425; PMCID: PMC10225218.
Cheng M, Geng C, Tang T, Song Z. Mucoepidermoid carcinoma of the breast: Four case reports and review of the literature. Medicine (Baltimore). 2017 Dec;96(51):e9385. doi: 10.1097/MD.0000000000009385. PMID: 29390541; PMCID: PMC5758243.
Thank you very much for your recommendations! We have performed the modifications and we think that the manuscript is much improved now!
Best regards,
Tiberiu et al.
Round 2
Reviewer 2 Report
Comments and Suggestions for Authors
The manuscript can be accepted in the present form